# Radiographic Assessment of CVC Malpositioning: How can AI best support clinicians?

**Lasse Hansen**[1]                                                        HANSEN@IMI.UNI-LUEBECK.DE
[1] *Institute of Medical Informatics, Universität zu Lübeck, Lübeck, Germany*

**Malte Sieren**[2]                                                        MALTE.SIEREN@UKSH.DE
[2] *Department of Radiology and Nuclear Medicine, UKSH Lübeck, Lübeck, Germany*

**Malte Hobe**[2]
**Axel Saalbach**[3]                                                        AXEL.SAALBACH@PHILIPS.COM
[3] *Philips Research Hamburg, Röntgentrasse 24-26, 22335 Hamburg, Germany*

**Heinrich Schulz**[3]                                                      HEINRICH.SCHULZ@PHILIPS.COM
**Jörg Barkhausen**[2]                                                      JOERG.BARKHAUSEN@UKSH.DE
**Mattias P. Heinrich**[1]                                                  HEINRICH@IMI.UNI-LUEBECK.DE

**Editors:** Under Review for MIDL 2021

## Abstract

The malpositioning of central venous catheters (CVCs) is a common technical complication that is usually diagnosed on post-procedure chest X-rays (CXRs). If the misplaced CVC remains undetected, it can lead to serious health consequences for the patient. Interpreting CXRs at a large scale in everyday clinical practice is time consuming and can create delays in the repositioning of the CVC. A computer-assisted assessment of post-procedure CXRs can help to prioritise cases and reduce human errors in stressful situations by objectifying decisions. However, final assessments must always be made by the clinicians, which is why the algorithmic support needs to be comprehensible.

Since AI models are not yet able to detect catheter maplpositons with highest accuracy, the focus must be on efficient support in everyday clinical practice. In this work, we evaluate three different AI models, particularly with regard to the relationship between classification accuracy and the degree of explainability. Our results show how helpful it is to incorporate explicit clinical knowledge into deep learning-based models and give us promising research directions for a planned large scale patient study.

**Keywords:** Chest X-Ray, CVC Malpositioning, Explainable AI

## 1. Introduction

If undetected, malpositioned central venous catheters (CVCs) can lead to permanent and severe damage for patients (Roldan and Paniagua, 2015). A verification on chest X-ray and possible repositioning of the catheter are standard clinical practice, but a prompt diagnosis at a large scale for all patients is time consuming and a computer assisted prioritisation of cases may be a significant benefit. Medical image analysis using artificial intelligence (AI, especially convolutional neural networks (CNNs)) allows diagnosis with high accuracy. However, final decisions have to be made by the clinicians which makes the explainability of algorithmic support as important as correct predictions. In this work, we propose a surprisingly intuitive model that combines deep learning based segmentation and localisation (of catheter, tip and target area) with an interpretable decision rule and outperforms direct classification for CVC positioning in chest X-rays.

## 2. Experiments and Results

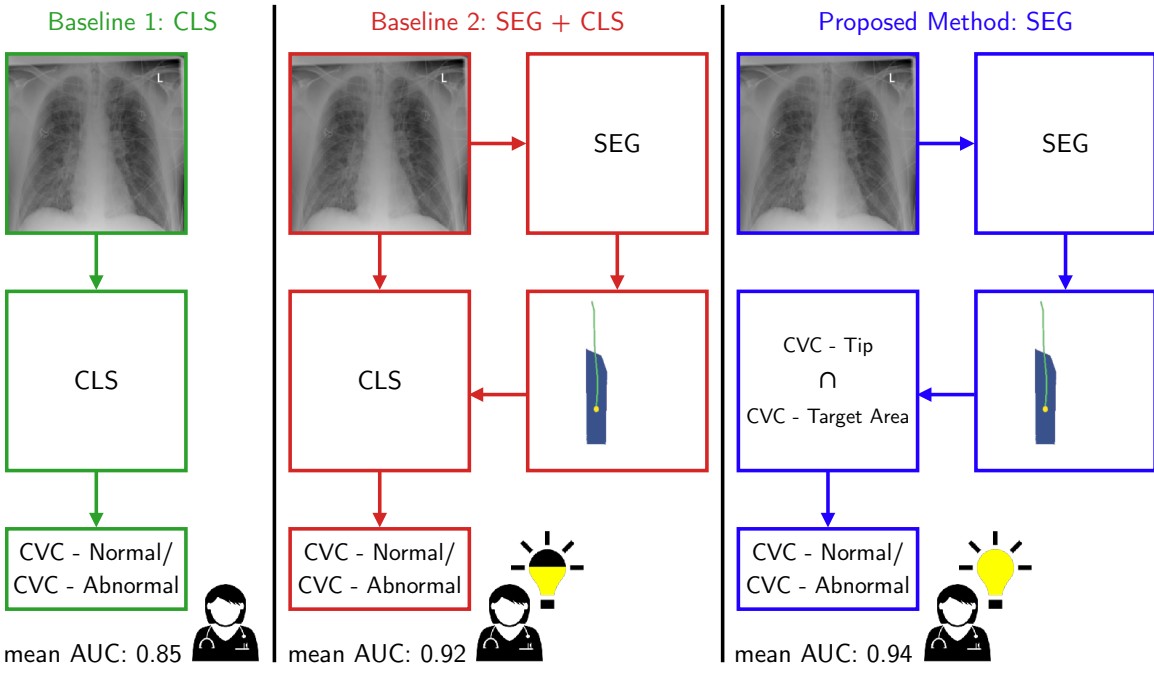

Figure 1: Overview of investigated methods for classifying CVC positioning in chest X-rays.

Figure 1 gives an overview of our three investigated baseline methods. Patients may have had several CVCs placed, which means we have multiple target labels for each chest X-ray image. **Baseline 1** directly predicts target labels using a classification network (CLS). For **Baseline 2**, in a first step segmentation labels (CVC - Tube ■ , CVC - Tip ■ , CVC - Target Area ■ ) are inferred using a segmentation network (SEG), which are then concatenated with the input image and processed with a classification network (CLS). Finally, we propose a third approach (**Proposed Method**), in which the soft intersection between the segmented CVCs tip and the target area is computed resulting in a score that can be directly used to decide whether the CVC is correctly positioned. We make use two datasets: 1) the publicly available **RANZCR CLiP** dataset[1], which contains ∼ 30.000 labeled chest X-ray images (including ∼ 9000 images with catheter segmentations) and 2) a dataset with 500 annotated (CVCs and target areas) chest X-rays from the Department of Radiology and Nuclear Medicine at the Universitätsklinikum Schleswig Holstein (**UKSH**) in Lübeck. Both datasets were used to train (finetune) the segmentation network (U-Net with EfficientNet-4 backbone (Ronneberger et al., 2015; Tan and Le, 2019)) while the classification network (EfficientNet-4) was trained solely with images from the RANZR CLiP dataset (we experienced overfitting when finetuning on the smaller UKSH dataset). Evaluation results (see Figure 1 and 2) are reported for the UKSH dataset (using a 5-fold

---

1. https://www.kaggle.com/c/ranzcr-clip-catheter-line-classification

cross validation for the segmentation task). With mean AUCs (Area under ROC curve) of 0.85, 0.92 and 0.94 for the two baselines and our proposed method the experiments clearly show the benefit of incorporating detailed segmentation labels in the classification process and indicate that our direct approach using soft intersections of the segmentation labels is competitive while having a much greater explanatory power.

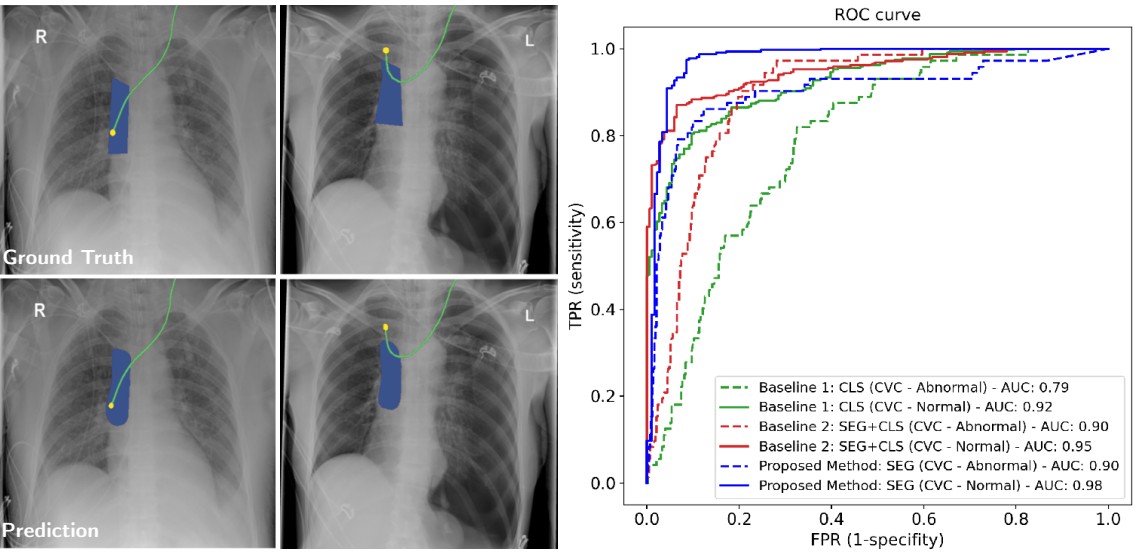

Figure 2: Qualitative and quantitative results of our experiments. On the left inferred segmentations (Tube ■ , Tip ■ , Target Area ■ ) are visualised, while the ROC curves on the right summarise classification results on the UKSH dataset.

## 3. Discussion and Outlook

Our results demonstrate that by combining a large-scale lower quality database (RANZR CLiP) with a smaller but more carefully curated set of annotations, we can substantially improve both CVC malpositioning classification as well as explainability of the results. Future work could comprehensively evaluate the influence and uncertainties of individual types of supervision and incorporate further anatomical guidance.

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
