# OpenReview forum: "Radiographic Assessment of CVC Malpositioning: How can AI best support clinicians?"
_MIDL.io/2021/Conference/Short — MIDL 2021 Poster_

### Official Review · Reviewer_pWvq · 2021-04-30

**Confidence:** 3
**Final Rating:** 3

**Summary:**

The paper compares three a deep-learning based methods for detecting malpositioned central venous catheters. Two of these methods rely on classic CNN-based classifiers, while a third only relies on a segmentation network. The resulting segmentation is used as input to an intuitive and interpretable rule-based decision. Furthermore, this interpretable method is shown to outperform black-box classification using CNN-based classifiers.

**Strengths:**

The problem is well introduced, highlighting the importance of interpretability of the method.

The proposed method is highly intuitive and presented clearly; figure 1 makes the differences between the three methods immediately obvious.

**Weaknesses:**

The evaluation seems somewhat misleading. Results are reported for the UKSH set, on which the segmentation method was fine-tuned, whereas the classification baselines were only trained on the RANZR CLiP set. This means it is possible that the difference in performance between the second baseline and the proposed method can be explained by domain shift between these two sets. The authors claim the classification network overfit when finetuning on the (smaller) test set folds, but it seems likely that performance would still improve when finetuning on this set and using techniques such as early-stopping.

**Deanonymize Review:**

yes

**Justification Of The Rating:**

The paper presents an elegant, interpretable method for central venous catheter malpositioning. The evaluation is a little questionable to me, as it presents results from competing methods where only the proposed method was trained with the evaluation dataset.

**Paper Type:**

validation/application paper

**Special Issue:**

no

---

### Meta-Review · Program_Chairs · 2021-05-10

**Recommendation:** Accept (Poster)
**Confidence:** 4

**Metareview:**

This paper is a well thought through application of deep learning to a clinically relevant problem. It is a nice example showing that not everything can be learned easily and specifically designed methods can still be have a clear advantage over more generic learning approaches, where in this case, a segmentation + rule-based approach works much better to detect catheter malpositioning than a more generic classification. In the final version it would be good to see, as the reviewer points out, also the results of classification with finetuning (or segmentation without finetuning) on the smaller dataset for a more fair comparison.

---

### Decision · Program_Chairs · 2021-05-11

Accept (Poster)